# Effect of Cannabidiol on Human Peripheral Blood Mononuclear Cells and CD4+ T Cells

**DOI:** 10.3390/ijms241914880

**Published:** 2023-10-04

**Authors:** Alessia Furgiuele, Franca Marino, Emanuela Rasini, Massimiliano Legnaro, Alessandra Luini, Maria Giulia Albizzati, Alessia di Flora, Barbara Pacchetti, Marco Cosentino

**Affiliations:** Center for Research in Medical Pharmacology, University of Insubria, 21100 Varese, Italy; alessia.furgiuele@uninsubria.it (A.F.); franca.marino@uninsubria.it (F.M.); emanuela.rasini@uninsubria.it (E.R.); massimiliano.legnaro@uninsubria.it (M.L.); alessandra.luini@uninsubria.it (A.L.); maria.albizzati@uninsubria.it (M.G.A.); a.diflora@uninsubria.it (A.d.F.); barbarapacchetti@gmail.com (B.P.)

**Keywords:** cannabidiol, TNF-α, IFN-γ, cell differentiation, cell proliferation

## Abstract

Cannabidiol (CBD), the main non-psychoactive component of *Cannabis sativa* L., is widely used in therapy for the treatment of different diseases and as an adjuvant drug. Our aim was to assess the effects of CBD on proinflammatory cytokine production and cell proliferation in human peripheral blood mononuclear cells (PBMCs) and on CD4+ T lymphocyte differentiation, and, furthermore, to test CBD’s ability to affect the functional properties of regulatory T cells (Treg). Experiments were performed on isolated PBMCs and purified CD4+ T lymphocytes obtained from the buffy coats of healthy subjects. Cytokines produced by CD4+ T cells were evaluated by flow cytometry and intracellular cytokine staining techniques. PBMC cytokine production was measured by an ELISA assay. Real-time PCR was used to assess the mRNA expression of cytokines and the key transcription factors (TFs) of CD4+ T cells. Finally, the proliferation of PBMC and CD4+ T effector cells (Teff), alone and in the presence of Treg, was assessed by flow cytometry. Results showed that CBD affects both the frequency of IL-4-producing CD4+ and of IFN-γ/IL-17-producing cells and dramatically decreases the mRNA levels of all TFs. Stimuli-induced cytokine mRNA expression was decreased while protein production was unaffected. CBD was unable to affect the ability of Treg to prevent Teff cell proliferation while it slightly increased PBMC proliferation. In conclusion, CBD may inhibit the expression of proinflammatory cytokines; however, the effect of CBD on cell proliferation suggests that this cannabinoid exerts a complex activity on human PBMCs and CD4+ T cells which deserves further investigation.

## 1. Introduction

Different cannabinoids can be extracted from the Cannabis sativa plant (*Cannabis sativa* L.). However, the main therapeutic relevance is ascribed to Δ9-tetrahydrocannabinol (Δ9-THC) and cannabidiol (CBD). The psychoactive properties of the plant are related to Δ9-THC, while CBD represents the main non-psychoactive constituent and seems of the highest interest for its clinical effects in the treatment of seizures associated with Lennox–Gastaut syndrome (LGS), Dravet syndrome, tuberous sclerosis complex (TSC), or chronic pain [1]. CBD and CBD-based products seem to be very promising in the treatment of a variety of CNS disorders and they are currently undergoing pre-clinical and clinical evaluation [2]. Moreover, several lines of evidence from a pre-clinical model of neuropathic pain, inflammatory-induced chronic pain, and arthritis-related pain indicate that CBD, and cannabinoids in general, could exert analgesic effects [3]. Indeed, cannabinoid receptors (CB1 and CB2) and their ligands are involved in the mechanisms of neuropathic pain [4]. At present, several studies strongly support the anti-inflammatory and immunomodulating properties of cannabinoids in different types of diseases such as inflammatory bowel disease, arthritis, vascular inflammation [5], and multiple sclerosis [6]. However, these studies were mainly conducted on animal models, while research in humans is still limited and fragmentary [6,7,8], as elegantly documented in a recent systematic review [9], despite the widespread use of cannabis and CBD-based products.

In particular, the mechanisms involved in the anti-inflammatory and immunomodulatory effects of cannabis products in humans have been only sporadically investigated. In this regard, we have previously shown that CBD inhibits some polymorphonuclear leukocyte (PMN) functions, including ROS production, migration, and tumour necrosis factor (TNF)-α production [10]. Moreover, CBD was able to inhibit COX-1 and COX-2 mRNA expression in activated PMN [11].

Among the immune cell-producing cytokines and chemokines, the products of lymphocytes, in particular, seem to play a crucial role in various autoimmune/inflammatory disorders. More in detail, CD4+ T lymphocytes play a crucial role in the orchestration of a successful immune response during the pathogenesis of inflammatory diseases as they may acquire proinflammatory phenotypes, such as T helper (Th) 1 and Th17, or, on the contrary, an anti-inflammatory phenotype, such as Th2 and T regulatory [12]. It is now well established that the master transcription factors (TFs) are necessary to drive Th differentiation and that TF dysregulation can contribute to tissue inflammation and the pathogenesis of several inflammation-related diseases [13]. Additional proof in this direction comes from the paper by Liu et al. that showed that a reduction in the transcriptional factor (TF) STAT3 levels in CD4 lymphocytes seems to inhibit the development of experimental autoimmune diseases [14].

According to all this evidence, we aimed to examine the effects of CBD on key immune responses, including CD4+ T lymphocyte differentiation and proliferation. To this end, we measured both the mRNA expression of TF and the intracellular cytokine content involved in the differentiation of CD4+ T cells. In addition, we measured the ability of CBD to affect both PBMC proliferation and cytokine release and the ability of Treg to counteract Teff proliferation.

## 2. Results

### 2.1. Effect of CBD on T Cell Functional Activation

#### 2.1.1. Flow Cytometric Evaluation of CD4+ T Cells Intracellular Cytokine Content

The flow cytofluorometric evaluation of intracellular cytokine content was very low; the percentage of resting cells positive for IFN-γ was 0.022 ± 0.019, 0.150 ± 0.071 for IL-17, and 0.004 ± 0.004 for IL-4. To better evaluate the ability of CBD to exert anti-inflammatory activities, we used as a positive control dexamethasone (DMX; 1 μM). The presence of CBD does not affect the % of cytokine-producing CD4+ T cells in the resting condition (*p* > 0.05). As expected, cell activation with PMA-IONO (see the methods section) induces an increase in the percentage of cytokine-producing CD4+ cells for all cytokines. CBD significantly affects the percentage of cells double positive for IFN-γ and IL-17 (Th1-Th17, Figure 1, panel d) and positive for IL-4 (Th2, Figure 1, panel b) that were reduced in comparison to the activated cells in the absence of CBD, while no differences were observed for the percentage of IL-17 (Th17) CD4+ T cells in comparison to activated cells alone (Figure 1, panel c). To better understand the effects of CBD, we can see that the effect of CBD on activated cells is superimposable to the effect exerted by DMX (always *p* > 0.05 for ACT + DMX vs. ACT + CBD), with the only exception of IL-4 (panel b); in this case, the effect of DMX on cell activation was more pronounced in comparison to the effect of CBD.

#### 2.1.2. TF mRNA Expression in Isolated Cultured PBMC

As shown in Figure 2, CBD alone was unable to affect the TF mRNA expression in resting PBMCs cultured for 48 h (Figure 2). As expected, cell activation with antiCD3/CD28 strongly increases all measured TFs (Figure 2). Treatment with CBD reverted the observed increase after cell stimulation for all TFs; in the majority of observations, the effect of CBD completely reverts the increase obtained by cell activation (except for GATA3 and STAT4). Interestingly in some cases, the reduction was so strong that no difference was observed between the values for the unstimulated cells (CBD alone) and stimulated cells in the presence of CBD (STAT1, STAT6, RORC, and STAT3).

### 2.2. Effect of Treatment with CBD on Cytokine mRNA Expression and Protein Production in Cultured Human PBMC

As shown in Figure 3, the resting mRNA expression of TNF-α, IFN-γ, and IL-17 was unaffected by the presence of CBD. As expected, cell stimulation with coated anti-CD3/soluble anti-CD28 significantly increased the mRNA expression of all three cytokines. Treatment with CBD decreased the effects of cell activation, even if the entity of the inhibitory effect was different depending on the cytokine analysed (Figure 3).

As shown in Table 1, cultured PBMCs in resting conditions were able to produce and release appreciable levels of TNF-α and IFN-γ, while the resting production of IL-17 remained always under the limit of detection (LOD); however, treatment with CBD did not affect resting cytokine production. As expected, stimulation induced a significant increase in all measured cytokines in comparison with the respective resting conditions, including IL-7. In this case, considering that resting levels were always lower than the LOD, the difference between the resting and stimulated conditions was calculated assuming resting values of 0. CBD never affected stimulus-induced cytokine production for all three cytokines analysed (Table 1).

### 2.3. Effect of Treatment with CBD on Cell Proliferation

As expected, PBMCs cultured for 120 h under stimulation with antiCD3/antiCD28 showed a dramatic increase in the percentage of proliferating cells (Table 2). We aimed to investigate if CBD was able to affect cell proliferation; to this end, we added CBD together with an activating stimulus, and, as shown in Table 2, treatment with CBD slightly but significantly increased cell proliferation in comparison with the stimulated cells alone (Table 2).

As for PBMCs, Teff proliferation significantly increased after a 120 h cell culture in the presence of the stimuli PHA/IL-2 (Table 3); the presence of Treg in cell culture completely prevents Teff cell proliferation. We aimed to investigate if CBD was able to modulate Treg inhibitory activity; to this end, CBD was added together with the coculture (Teff/Treg 1:1) in the presence of activating stimuli. As shown in Table 3, CBD was unable to affect Treg inhibitory ability (Table 3).

## 3. Discussion

The main result of this study is that CBD can modulate CD4+ T cell differentiation and cytokine mRNA expression in human PBMCs. Moreover, it increases the cell proliferation of PBMCs without affecting the inhibitory ability of Treg to prevent Teff proliferation. By means of intracellular cytokine staining and flow cytometry, we have shown that CBD, in activated cells, affects the frequency of all the cytokines investigated, with the only exception being the Th1 cytokine IFN-γ. Interestingly, the effect of CBD was superimposable to what was observed with DMX, a well-known anti-inflammatory and immunosuppressant agent. In fact, CBD and DMX show a similar profile on active cells, with the exception of the IL-4 content; in this case, the effect observed with DMX was more pronounced in comparison to CBD. Additionally, CBD profoundly reduced, although to a different extent, all the TF expression during cell activation without affecting the resting mRNA expression. Similarly, we showed that CBD counteracts the stimuli-induced mRNA expression of IFN-γ, TNF-α, and IL-17 in cultured PBMCs; however, this effect was not replicated in protein production.

In this study, we tested CBD at a fixed concentration of 1 µM which is usually achieved in humans with doses of the CBD drug Epidiolex^®^, approved for the treatment of drug-resistant seizures in Lennox–Gastaut syndrome or Dravet syndrome [15].

It seems that the modulation of arachidonic acid release, peroxisome proliferator activated-receptor-γ (PPAR-γ), reactive oxygen intermediate (ROI) production, cytokines, and the cyclooxygenase pathway [16,17] can contribute to the anti-inflammatory actions of CBD. In line with this evidence, recently, we have found that CBD profoundly affects some key human neutrophil functions, showing a powerful anti-inflammatory profile [10,11]. In addition, it was recently shown that the exogenous cannabinoid WIN55212-2 is able to affect the T cell response [18] or modulate the functions of monocytes and macrophages inhibiting LPS-induced inflammation [19], corroborating our and other studies suggesting that cannabinoids, through complex and varied mechanisms, can modulate the immune response and, therefore, probably be used for different therapeutic purposes in addition to what at present is sustained.

The key role played by the adaptive immune system, and, in particular, by CD4+ T helper (Th) lymphocytes, in inflammation-driven neurodegenerative diseases such as multiple sclerosis and Parkinson’s disease was widely investigated [20]. We established that CBD modulates CD4+ T cell differentiation. The evaluation of the intracellular cytokine content by flow cytometry showed that cell activation increases the percentage of CD4+ producing IFN-γ, IL-4, and IL-17. However, CBD significantly reduces both the percentage of cells double positive for IFN-γ and IL-17A (Th1-Th17) and the percentage of cells positive for IL-4 (Th2). The role played by IFN-γ and IL-17 in inflammatory diseases is well known [20]; however, the data on IL-4 are not so easy to understand when considering that this cytokine is often associated with anti-inflammatory activity, although data in this regard are controversial [21].

PBMC activation with anti-CD3/anti-CD28 induces an increase in all measured TFs and CBD significantly reduces the effect of cell stimulation on their expression. The TFs measured have different physiological roles, in particular, TBX21, STAT4, and STAT1 drive Th1 differentiation [22], while RORC and STAT3 regulate the differentiation towards the Th17 lineage [23]; STAT6 and GATA3 control Th2 development [24,25]; FOXP3 is a key regulator for the development and maintenance of the Treg phenotype [26]; and NR4A2 represents the encoding of the orphan nuclear receptor Nurr1, which affects Treg cell development through the activation of FOXP3 [27]. We have previously shown that TFs can be differently regulated in several kinds of diseases and can be considered predictive for disease progression [28,29]. It must be noted that we used real-time PCR to measure the mRNA levels of TFs, which is not yet a measure for gene transcript activity; however, the modification of TF mRNA levels can be considered an indicator of their involvement in the above-mentioned processes. The observation that CBD significantly reduces all TF mRNA expression can be considered as the ability of the product to affect T cell functions and this issue deserves a more deepened investigation, perhaps investigating the intracellular pathways activated. In any case, these data are in line with the above-mentioned possible other clinical use of this safe product, for example, in slowing the inflammation of neurodegenerative diseases.

The clinical use of drugs containing cannabis derivatives for the treatment of different CNS diseases is supported by several studies [2]; CBD, and cannabinoids in general, could exert analgesic and anti-inflammatory effects in several pain-related diseases [2,3,4]. The ability of CBD to trigger positive effects in several diseases was widely investigated and described [1,5,6,10,30], but, recently, it was also shown that cannabis smoking is able to induce changes in the expression of DNA-methyltransferase in lymphocytes, even if these results are not clearly related to specific clinical relevance [31].

In 2005, the Food and Drug Administration authorised *nabiximol* (Sativex^®^), a mixture of Δ-9-THC and CBD (1:1), for the treatment of pain and spasticity in MS [32] and *cannador*, used to treat pain in the same disease [33]. Moreover, although not extensively investigated in humans, *epidiolex* is an oil formulation of CBD that has been used in clinical trials for the treatment of neuropathic pain and inflammation in different neurological diseases [34,35].

Interestingly, the final effect of CBD seems to depend on the range of the used concentrations. Watzl and colleagues showed that 10–100 ng/mL of CBD increased IFN-γ and TNF-α production in mitogen-stimulated PBMCs, while higher concentrations (higher than 60–100%) completely blocked this production [36], suggesting the effects exerted by CBD on the immune system to be complex, thus deserving a better and deeper investigation of the intracellular mechanism involved in said effects.

## 4. Materials and Methods

### 4.1. Substances

CBD powder (white/off-white or slightly yellow powder, batch no EP20/054-07) was kindly provided by EMMAC, Switzerland (now CURALEAFINT Curaleaf International Limited, London, UK) and dissolved in dimethylsulfoxide (DMSO; Sigma-Aldrich, St. Louis, MI, USA). Stock solutions (0.1 mM) were stored at −80 °C until use. Daily fresh solutions were prepared in RPMI 1640 medium (Euroclone ECM0495L). The final concentration of DMSO in the cell culture was 0.001%, which in previous studies was shown to be non-cytotoxic [9].

### 4.2. Peripheral Blood Mononuclear Cell (PBMC) Isolation and Functional Assays

Human peripheral blood mononuclear cells (PBMCs) were obtained from unemployed blood buffy coats provided by the local blood bank (Ospedale di Circolo, Fondazione Macchi, Varese, Italy); these materials represent a discard of the whole blood for clinical use and therefore no Ethical Committee approval is required. Cells were separated by means of a FICOLL density gradient centrifugation (a detailed description of this procedure is available at: http://dx.doi.org/10.17504/protocols.io.bpxjmpkn (accessed on 6 August 2023)). Briefly, cells were resuspended, and any contaminating erythrocytes were lysed by adding 5 mL of lysis buffer ((g/L) NH4Cl 8.25, KHCO3 1.0, EDTA 0.037); then, cells were incubated under continuous gentle vortexing for 5 min and centrifugated at 100 g for 10 min at room temperature (RT). Finally, the cells were washed twice in 15 mL of PBS and centrifuged at 300 g, 10 min at RT; cells were resuspended at the desired final concentration in RPMI/10% FBS for subsequent culture. For each experimental set, only PBMCs that presented a composition of at least 80% lymphocytes (assessed by flow cytometry) and viability, assessed by the trypan blue exclusion test higher or equal to >99%, were used, as better detailed in: http://dx.doi.org/10.17504/protocols.io.bpxtmpnn (accessed on 6 August 2023)).

#### 4.2.1. PBMC Proliferation Assay

Freshly isolated PBMCs were resuspended at 1 × 10^6^ cells/mL in a complete culture medium (RPMI/FBS 10%/Pen-Strep 100 U/mL). Briefly, the medium composition was: RPMI 1640 medium supplemented with 10% heat-inactivated FCS, 2 mM glutamine, and 100 U/mL penicillin/streptomycin. Cells were incubated at 37 °C in a moist atmosphere of 5% CO_2_; more detail at http://dx.doi.org/10.17504/protocols.io.byvapw2e (accessed on 6 August 2023). The cell suspension (250 μL/well) was placed in a 96-well flat bottom plate alone or the presence of CBD 1 µM and cultured at 37 °C in a moist atmosphere of 5% CO_2_. Cells were left in resting conditions or stimulated with a plate-bound anti-CD3 human antibody (clone UCHT1, BD Biosciences 555330) and a soluble anti-CD28 human antibody (clone CD28.2, BD Biosciences 555726), each used at a concentration of 2 μg/mL. PBMC proliferation was measured after 120 h of cell culture by staining cells with the cell proliferation dye eFluor670 2.5 µM (CPD670, eBioscience-Life Technologies Italia), according to the standardised protocol, and samples were analysed by flow cytometry as described below. The production of cytokines TNF-α, IFN-γ, and IL-17A was assessed in supernatants, which were collected after 48 h of cell culture and frozen at −80 °C until analysis by means of ELISA assays, as described below. Cell pellets from the same culture were used to evaluate the mRNA expression levels of both cytokines (TNF-α, IFN-γ, and IL-17A) and transcription factor (TF) genes TBX21, STAT1, STAT3, STAT4, STAT6, RORC, GATA3, FOXP3, and NR4A2 by means of a real-time polymerase chain reaction (RT-PCR) as described below.

#### 4.2.2. PBMC Cytokine Production

For cytokine production (TNF-α, IFN-γ, and IL-17A), PBMCs were stimulated for 48 h with coated anti-CD3/soluble anti-CD28 (2/2 μg/mL). At the end of the culture, the supernatant and cells were collected, stored, and frozen at −80 °C until analysis. Cytokine production was assessed in supernatants by means of the ELISA technique and in cell pellets by means of real-time PCR for mRNA expression.

#### 4.2.3. Transcriptional Factors’ mRNA Expression in PBMCs

Cells were cultured as described for cytokine production. At the end of the culture, the cell pellets were removed and frozen until the assay was performed to analyse the transcription factor (TF) genes TBX21, STAT1, STAT3, STAT4, STAT6, RORC, GATA3, FOXP3, and NR4A2 by means of the real-time polymerase chain reaction (RT-PCR) as described below.

### 4.3. Isolation and Proliferation Assay of T Regulatory and T Effector CD4+ T Lymphocytes

Effector T lymphocytes (Teff) and regulatory T lymphocytes (Treg) were isolated from PBMCs using the human CD4+CD25+ Regulatory T cell Isolation Kit (code 130-091-301, Miltenyi Biotec) according to the manufacturer’s instructions. In detail, according to our previous work [37], Tregs and Teffs were isolated from PBMCs (that were isolated according to the above method) by immunomagnetic sorting using the Dynal CD4 CD25 Treg Kit (Dynal, Oslo, Norway), according to the manufacturer’s instructions. Free-beeds Tregs and Teffs (as assessed by electron microscopy) were analysed for purity and viability; only cells with a purity higher or equal to 95% (assessed by flow cytometry) and vitality more than 99% (Trypan Blue exclusion test) were used. For the subsequent culture, cells were resuspended in RPMI 1640 medium supplemented with 10% heat-inactivated FCS, 2 mM glutamine, and 100 U/mL penicillin/streptomycin and incubated at 37 °C in a moist atmosphere of 5% CO_2_, as detailed in the published online protocol (dx.doi.org/10.17504/protocols.io.bpxumpnw (accessed on 6 August 2023)). Isolated Teff was cultured alone or co-culture with Treg (ratio Teff:Treg 1:1) in 96-well plates at a concentration of 1 × 10^6^/mL in a complete medium. According to our previous study [28], Teff/Treg stimulation to measure cell proliferation was performed using a cocktail of phorbol myristate acetate (PHA)/interleukin (IL)-2; this represented a better stimulus for our model instead of the antiCD3/antiCD28 used to measure PMBC proliferation. Teff or Teff/Treg cells were treated with CBD (1 μM) and stimulated with PHA (5 μg/mL) and IL-2 (40 ng/mL). Cell proliferation was evaluated after a 120 h cell culture by staining Teff with CPD670 2.5 µM and the samples were analysed by flow cytometry as described below.

### 4.4. PBMC Stimulation and Intracellular Cytokine Staining of IFN-γ-, IL-4-, and IL-17A- Producing CD4+ T Cells

Freshly isolated PBMCs were cultured and stimulated according to the following procedures and better detailed in the protocol published online (http://dx.doi.org/10.17504/protocols.io.byrppv5n (accessed on 6 August 2023)). Cells were resuspended at 8 × 10^6^ cells/mL in complete medium (medium composition: RPMI/FBS 10%/Pen-Strep 100 U/mL) with phenol-red supplemented with 5 mL of FBS (foetal bovine serum). Then, 0.5 mL penicillin/streptomycin (equivalent to 100 U/mL) and 250 μL (2 × 10^6^ cells) were placed in a 96-well flat bottom plate. PBMCs were cultured for 5 h alone or treated with CBD 1 µM in unstimulated conditions or added with 10 ng/mL phorbol 12-myristate 13-acetate (PMA, Sigma-Aldrich, Italy) and 1 µg/mL ionomycin (Sigma-Aldrich, Italy) in the presence of 10 μg/mL of the protein secretion inhibitor brefeldin A (Sigma-Aldrich, Italy). DMX (1 µM) was used as a positive control and added together with the activating stimuli. Cells were then fixed/washed with BD Cytofix and BD Perm/Wash buffers (Becton Dickinson, Milano, Italy) and stained with a cocktail of anti-human CD4-PerCP/Cy 5.5, IFN-γ-FITC, IL-17A-PE, and IL-4-APC antibodies for 30 min at RT in the dark (antibodies and volumes used are listed in the Appendix A). Finally, samples were washed with 1 mL Perm/Wash buffer, centrifuged at 1200× *g* for 5 min, and suspended in 400 µL PBS/FBS 2% prior to flow cytometric analysis.

### 4.5. RNA Isolation and Real-Time Polymerase Chain Reaction

Cytokine and TF mRNA expression were evaluated as previously described [28]. Briefly, PBMCs (5 × 10^4^) were resuspended in Perfect Pure RNA lysis buffer (5 Prime GmbH, Hamburg, Germany), and the total RNA was extracted by a PerfectPure RNA Cell Kit™ (5 Prime GmbH). The amount of extracted RNA was estimated by spectrophotometry at λ = 260 nm and reverse-transcribed using a random primer and a high-capacity cDNA RT kit (Applied Biosystems). cDNA amplification was performed by means of the SsoAdvanced™ Universal Probes Supermix (BIORAD) and assayed on a StepOne^®^ System (Applied Biosystems). The linearity of the assays was tested by constructing standard curves using serial 10-fold dilutions of a standard calibrator cDNA for each gene. Regression coefficients (r2) were always >0.999. Assays were performed in triplicate for each sample, and levels of mRNA were finally expressed as 2^−ΔCt^, where ΔCt = [Ct (sample) − Ct (housekeeping gene)]. The relative expression was determined by normalisation to the expression of RPS18 (gene for 18S cDNA). Data analysis was performed by StepOne software™ 2.2.2 (Applied Biosystems, Waltham, MA. USA). The primers used are listed in Appendix A.

### 4.6. Cytokine Analysis by ELISA

The production of TNF-α, IFN-γ, and IL-17A was measured in culture supernatants using commercial ELISA kits and following the manufacturer’s supplied protocols. ELISA kits were all purchased from Thermo Fisher Scientific (Invitrogen, Milan, Italy) and the cytokine content in the culture medium was measured using a Multiskan™ FC Microplate Photometer (Thermo Fisher Scientific, North Ryde, Australia). Finally, the data were analysed according to the manufacturer’s instructions and expressed as pg/mL.

### 4.7. Flow Cytometry

*Proliferation*. At the end of a 120 h cell culture, samples were washed with 1 mL PBS and acquired on a BD FACSCelesta flow cytometer (Becton Dickinson, Milan, Italy) with BD FACSDiva software (version 8.0.1.1). Data were analysed using FlowJo software (version 10.8.1). PBMCs or Teff cells stained with CPD670 were identified and gated according to their morphological parameters on a Forward Scatter (FSC) vs. Side Scatter (SSC) bi-parametric dot plot, the usual analysis technique employed to identify the cells positive for the staining used. At least 20,000 cells/sample were collected at the gate. The resting and activated cells were then visualised on a single-parameter histogram of the CPD signal, and proliferation was quantified as the percentage (%) of CPD+ cells with a reduced CPD fluorescence intensity following the dye dilution in daughter cells.

*Intracellular cytokine expression in CD4+ T cells*. The acquisition was performed by a BD FACS Celesta flow cytometer (Becton Dickinson Italy, Milan, Italy) using BD FACSDiva software (version 8.0.1.1). The gating strategy used to identify IFN-γ^+^ (Th1), IL-4^+^ (Th2), IL-17A^+^ (Th17), and IFN-γ^+^/IL17A^+^ (Th1-Th17) cells among CD4^+^ T lymphocytes is shown in Appendix A.

### 4.8. Statistics

Data were reported as the mean ± standard deviation (SD) or the mean ± SD and 25th–75th percentile, and n, with n indicating the number of observations. The statistical significance for correlations was set at *p* < 0.05 and calculated by means of the Student *t*-test for paired or unpaired data, as appropriate. Calculations were performed using commercial software (GraphPad Prism version 9.00 GraphPad Software, San Diego, CA, USA, www.graphpad.com).

## 5. Conclusions

In conclusion, these results show that CBD is able to affect the differentiation and some pivotal functions, such as the mRNA expression for TFs involved in cell activation and intracellular cytokine contents, in human cultured lymphocytes, suggesting that this compound can have potential applications as an anti-inflammatory agent. Thus, these preliminary results encourage further investigation of the potential anti-inflammatory and analgesic effects of CBD. According to data from the literature, our data on the reduction in cytokines (in the protein content of Th1/Th17 activated cells and mRNA expression of all cytokines) encourage additional studies on the potential role of CBD alone or in combination with other drugs to treat, for example, inflammation-related pain. The possible clinical implication of such a strategy is clearly evident considering that, in this view, it is possible to realise an optimal result with a lower dose of both drug classes, finally achieving a reduction in side effects and an increase in quality of life in the treatment of several immune-mediated diseases.

## Figures and Tables

**Figure 1 ijms-24-14880-f001:**
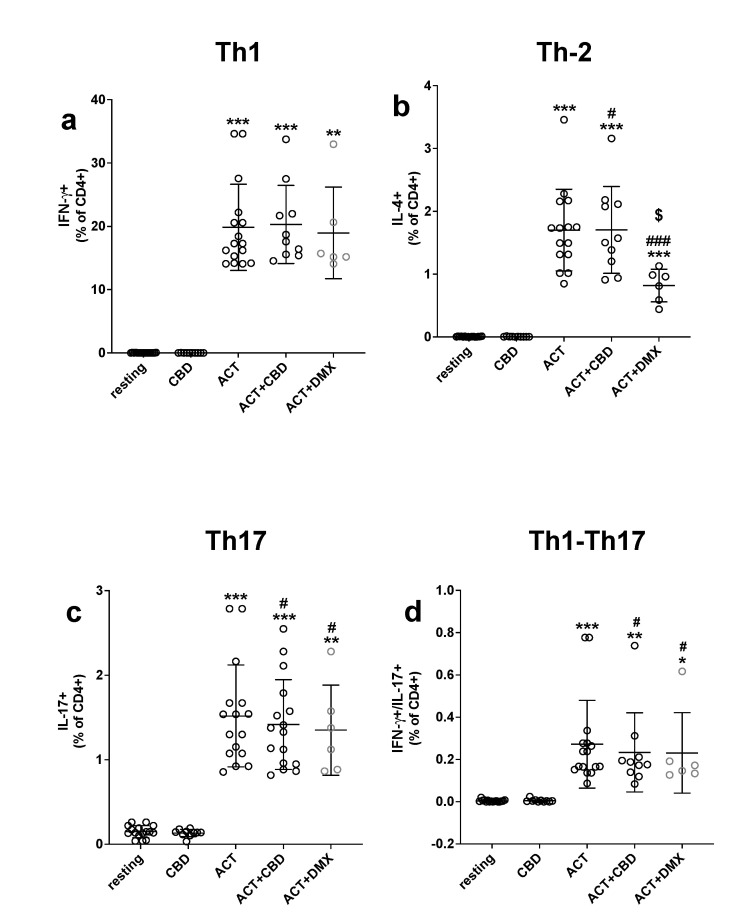
Effect of CBD treatment on intracellular cytokine production. PBMC were stimulated with 10 ng/mL phorbol 12-myristate 13-acetate (PMA) and 1 µg/mL ionomycin (IONO) in the presence of 10 μg/mL of the protein secretion inhibitor brefeldin A and put in culture for 5 h as described in the method section. CBD (1 μM) and DMX (1 μM) were added simultaneously with the stimuli. Cytokine-producing CD4+ T cells were measured by flow cytometry and the gating strategy shown in Appendix A. Panel (**a**) shows the effect of CBD on Th1 cytokine, panel (**b**) on Th2, panel (**c**) on Th17 while panel (**d**) shows the effect of CBD on Th1-Th17 ratio. Data are expressed as the mean (horizontal bars in the graph) of a single experiment (a circle for each experiment; 6–16 separate experiments) ± SD (vertical bars in the graph). The analysis for each group was performed by means of a one-way analysis of variance with Dunn’s post-test. * = *p* < 0.001 and ** = *p* < 0.0001, *** = *p* < 0.00001 vs. respective resting conditions; # = *p* < 0.05, ### = *p* < 0.0005 CBD or DMX vs. activated alone; $ = *p* < 0.05 vs. ACT + CBD.

**Figure 2 ijms-24-14880-f002:**
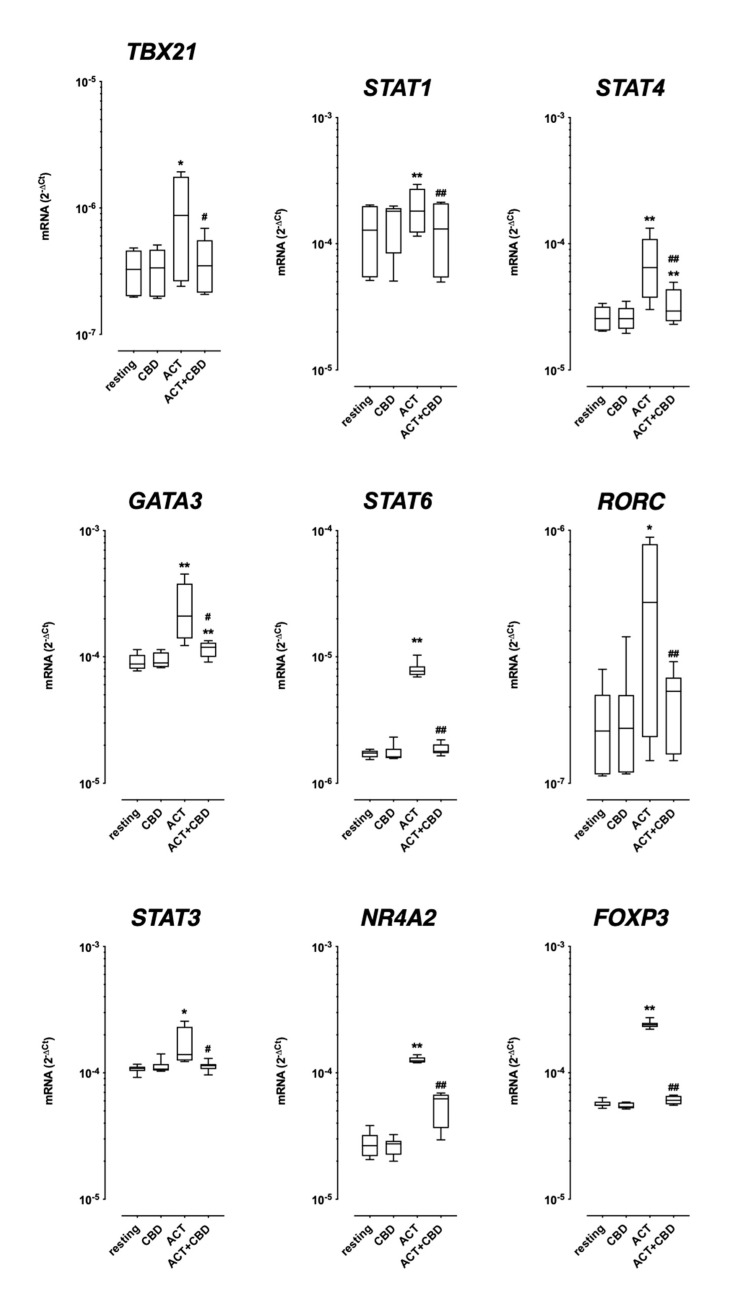
Effect of CBD treatment on the mRNA expression of transcriptional factors. PBMCs were cultured and activated (ACT) for 48 h as described in the method section, and CBD was added simultaneously with cell activation. The mRNA expression levels of TFs were measured by RT-PCR as described. Data are expressed as the median ± 25th–75th percentile of eight separate experiments. The analysis for each group was performed by means of paired Student *t*-test * = *p* < 0.001 and ** = *p* < 0.0001 vs. respective resting conditions. # = *p* < 0.001 and ## = *p* < 0.0001 vs. activated alone.

**Figure 3 ijms-24-14880-f003:**
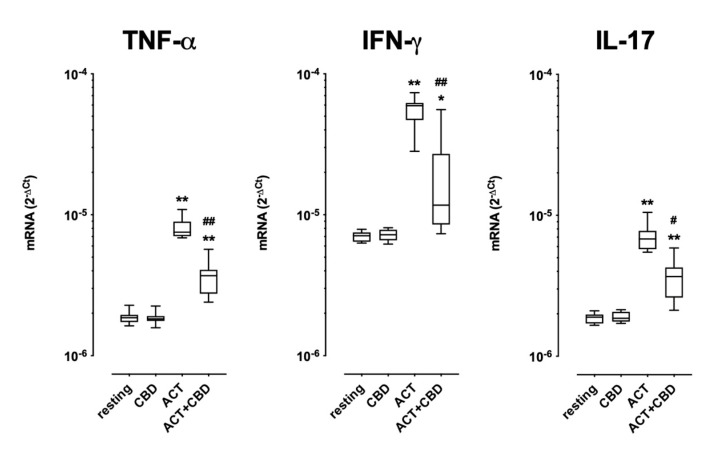
Effect of CBD on cytokine mRNA expression. Effect of CBD on mRNA expression of TNF-α (left), IFN-γ (middle), and IL-17 (right) in cultured PBMCs under resting and activated conditions for 48 h as described in the method section. CBD was added together with the cell activation. Data are presented as the median ± 25th–75th percentile. Analysis for each group was performed by means of paired Student *t*-test * = *p* < 0.001, ** = *p* < 0.0001 vs. resting conditions; # = *p* < 0.001, ## = *p* < 0.0001 vs. activated conditions.

**Table 1 ijms-24-14880-t001:** Effect of incubation with CBD on cytokine production. PBMCs (1 × 10^6^ cells/mL) were cultured for 48 h under resting conditions or stimulated with coated anti-CD3/soluble anti-CD28 (2/2 μg/mL). CBD (1 μM) was added together with cell activation. Data are pg/mL and are expressed as the mean ± SD of 10 separate experiments. Limit of detection (LOD). The analysis for each group was performed by means of a paired Student *t*-test * = *t*-test was performed assuming respective resting values of 0.

	Resting	Resting + CBD	Activated	P vs. Resting	Activated + CBD	P vs. Activated
TNF-α	24.3 ± 16.6	39.9 ± 23.9	2010.4 ± 1176.7	0.0005	2437.1 ± 1555.7	ns
IFN-γ	5.9 ± 2.4	5.5 ± 2.4	1251.3 ± 1631.8	0.039	1088.2 ± 1521.9	ns
IL-17	Under LOD	Under LOD	36.8 ± 24.4	0.04 *	33.6 ± 24.6	ns

**Table 2 ijms-24-14880-t002:** Effect of CBD on PBMC proliferation. PBMCs (1 × 10^6^ cells/mL) were cultured for 120 h under resting conditions or stimulated with coated anti-CD3/soluble anti-CD28 (2/2 μg/mL) (ACT). CBD (1 μM) was added together with cell activation. Data are expressed as the percentage of positive cells and presented as the mean ± SD of five separate experiments. The analysis for each group was performed by means of a paired Student *t*-test. Activated with coated anti-CD3/soluble anti-CD28 = ACT, resting = R.

R	CBD	P vs. R	ACT	P vs. Resting	ACT + CBD	P vs. ACT Alone
0.56 ± 0.30	0.78 ± 0.48	ns	50.90 ± 22.80	<0.01	60.40 ± 21.20	<0.05

**Table 3 ijms-24-14880-t003:** Effect of CBD on Teff/Treg proliferation assay. Teff/Treg (ratio 1:1) were cultured for 120 h under resting or activated (ACT) conditions as described in the method section. Cells were activated with a cocktail of phorbol myristate acetate (PHA; 5 μg/mL) and interleukin (IL)-2 (40 ng/mL). CBD (1 μM) was added together with cell activation (PHA/IL-2). Data are expressed as the percentage of positive cells and presented as the mean ± SD of seven separate experiments. The analysis for each group was performed by means of a paired Student *t*-test. Activated with PHA/IL-2 = ACT; not significant = ns; and resting = R.

R	CBD	P vs. R	ACT Teff	P vs. R	ACT Teff/Treg	P vs. ACT Alone	ACT Teff/Treg + CBD	P vs. ACT Teff/Treg
0.60 ± 0.20	0.58 ± 0.50	ns	71.40 ± 6.01	<0.01	5.23 ± 2.38	<0.001	6.67 ± 2.89	ns

## Data Availability

All collected data for this study will be made available upon request; all applied protocols are also available.

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
