# Peer review of "Effect of Cannabidiol on Human Peripheral Blood Mononuclear Cells and CD4+ T Cells"

_ijms, 2023, doi:10.3390/ijms241914880_

Round 1

Reviewer 1 Report

I would like to congratulate the author on such a good job, this article is logical and focused. However, before acceptance for possible publication, I still ask the following questions to help the author improve:

1. Is it possible to increase the comparison between your study and the results of other similar studies in appropriate sections, such as the introduction? This will better highlight your research contributions.

2. The name of Figure 1/2/3 is too long; it is recommended to leave only the backbone part and describe the rest in footnotes.

3. The charts in the appendix are suggested to be integrated into the main body, as most MDPI articles do.

4. The references are less recent, with only one reference for 2023, suggesting an appropriate increase.

Author Response

I would like to congratulate the author on such a good job, this article is logical and focused. However, before acceptance for possible publication, I still ask the following questions to help the author improve:

Many thanks for the positive comment’s

Hereafter the answer’s to your questions:

  1. Is it possible to increase the comparison between your study and the results of other similar studies in appropriate sections, such as the introduction? This will better highlight your research contributions.

In the Introduction section a recent systematic review [Rodriguez Mesa et al., 2021] summarizing the recent results in this topic was added

  1. The name of Figure 1/2/3 is too long; it is recommended to leave only the backbone part and describe the rest in footnotes.

Figure legends were revised

  1. The charts in the appendix are suggested to be integrated into the main body, as most MDPI articles do.

Revised

  1. The references are less recent, with only one reference for 2023, suggesting an appropriate increase.

References in the Introduction and Discussion sections were increased

Reviewer 2 Report

In this work Furgiuele and colleagues explored the effect of a major  cannabis constituent (CBD) on human PBMCs and CD4+ cells, by complementary techniques. However, several aspects must be discussed in order to consider the present research suitable for acceptance.

The authors  declare (L19) that CBD  affects the frequency of IL-4 producing CD4+ cells  they must avoid ambiguous terms and clearly define  the affectation.  The data from Figure 1, upper, right,   does not support this affirmation, as CBD and resting cells exhibit incipient levels of IL4+ production.  If the authors  refer to the effect of CBD in activated cells,  where affectation is supposedly observed, the authors may discuss the significance of such change. E.g., approximately 1.9% of il4 producing CD4+ cells  vs 1.7% of CBD treated cells.  Moreover, considering the great deviation of the presented data, is hard to seriously consider these conditions significantly different by the statistical test, which must be stated in the figure legend. If the authors don’t mind, it is of my particular interest to confirm this data.

Same for L19/20 stating that Results show that CBD affects the frequency o of IFN- g/IL-17-producing cells . what is the biological significance of such an small change (e.g., 0.27% to 0.22%?) and with such a big deviation.

It is somehow confusing why does the authors  activate some times with PMA or Ionomycin, then with anticd3/Cd28, some times they use CD4+ cells and some times PBMCs,  some times the activation are in a period of 5 h  in other experiments 48 h… this must be clearly explained in other to compare or discuss the obtained results.

I personally found the data not clear and superficial, the authors must want to discussed or explore in more detail the mechanisms and implications of the observed effects.

Additionally:

Title: Peirpheral / nd

Keywords         it is highly encouraged to restrict the keywords to the TNF and IFN evaluated in this research.

22        Teff must be defined     

23        Conclusions must be determined with clarity.  The authors must avoid the use of conditional terms like “may”  to conclude the effects of CBD in the cytokines production.  Additionally the data does not clearly support this conclusions (Table 1).

50        PMN must be defined

73        2.1. Effect of CBD on T cells differentiation . This title does not correspond to the cells employed in this section.

77/124  informal terms must be avoided in scientific texts. Please do not use contractions

77/78    and so on… % > percentage

79        figure> Figure, and must be bolded

80        PMA and IONO acronyms must be stated

82        only Figure number bolded, the rest must be normal text.

Figure 1           in legend, concentrations of PMA and ionomycin must be stated here

Figure 1           figure content must be indicated as a, b, c ,d  same for other figures.

Figure 1           please switch the column bars to scatter dot plot, it would be more informative. I personally consider it hard to appreciate the significance of the statistical approach employed with such a small change between conditions, and enormous deviations. Also, the statistical test must be stated in figure legend.

Figure 1           for TH1-Th17 please clearly state that # is employed to compare ACT-CBD vs ACT alone. If it is the case is very hard to consider this data significantly different. Please state the statistical test employed.

Figure 2           Please state the statistical test employed.

Section 2.3       the data described here must be presented as figure or table with corresponding annotations (methodological, statistical…)

Figure sup. 1    what does stimulated mean? Authors must state clearly the methods employed.

201      citations must be homogenized to the journal format

216      0,1 > 0.1

Materials and methods  authors must include a section attending to the following instruction: when reporting on research that involves human subjects, human material, human tissues, or human data, authors must declare that the investigations were carried out following the rules of the Declaration of Helsinki of 1975 (https://www.wma.net/what-we-do/medical-ethics/declaration-of-helsinki/), revised in 2013. According to point 23 of this declaration, an approval from the local institutional review board (IRB) or other appropriate ethics committee must be obtained before undertaking the research to confirm the study meets national and international guidelines. As a minimum, a statement including the project identification code, date of approval, and name of the ethics committee or institutional review board must be stated in Section ‘Institutional Review Board Statement’ of the article

Materials and methods  the authors must clearly described the cells culture conditions. Media composition, etc.

Materials and methods.             It is allowed to briefly describe well-known methods, but it has to be appropriately cited, however in my opinion, the proposed citations employed by the authors (links) to explain the methods are not friendly or appropriated, as readers have to open at least 6 independent web pages to understand this section. Anyhow, this section should be described with sufficient detail to allow others to replicate and build on published results.

238 Technologies Italia) > Technologies, Italia)

271      perm/whash

336      CD4 positive cells (CD4+) it must be described since the first time of appearance, not here

344/345 is CD4+ meaning was defined previously it is not necessary to write CD4+ positive

335      the authors must explain why is it necessary to abbreviate lymphocytes here? If it is not employed anywhere…

304      FSC and SSC the authors must state the meaning of this parameters, as the term is only informative for cytometrist.

289 Suppl Table 2  > Suppl.

344 homogenize the table name to Suppl. Table 1 and homogenize font size or  type

writing issues were detected and indicated in the comments for authors.

Author Response

Many thanks for your comments and we are grateful for the suggestion’s; the manuscript in our opinion is very improved and the data showing the ability of CBD to affects human PBMCs and CD4+ cells function's are more clear.

Hereafter the answer’s to your questions:

In this work Furgiuele and colleagues explored the effect of a major  cannabis constituent (CBD) on human PBMCs and CD4+ cells, by complementary techniques. However, several aspects must be discussed in order to consider the present research suitable for acceptance.

The authors  declare (L19) that CBD  affects the frequency of IL-4 producing CD4+ cells  they must avoid ambiguous terms and clearly define  the affectation.  The data from Figure 1, upper, right,   does not support this affirmation, as CBD and resting cells exhibit incipient levels of IL4+ production.  If the authors  refer to the effect of CBD in activated cells,  where affectation is supposedly observed, the authors may discuss the significance of such change. E.g., approximately 1.9% of il4 producing CD4+ cells  vs 1.7% of CBD treated cells.  Moreover, considering the great deviation of the presented data, is hard to seriously consider these conditions significantly different by the statistical test, which must be stated in the figure legend. If the authors don’t mind, it is of my particular interest to confirm this data.

We agree with your consideration. The data about the effects of CBD on resting ICS were a refused of a previous analysis. In this revised version, no differences was found between resting values alone and CBD. While, considering the data about the effects of CBD on activated cells, at least in this model, also little variation seems to reflect the effects of drugs on ICS content. As shown in the present version of the figure and discussed in the result section, in our model, we used as positive control for each cytokine evaluation, dexamethasone (DMX in the figure), that is a well-known anti-inflammatory and immunosuppressant drug. In our previous version, we don’t add the effect of DMX considering that is was not the focus of our study; in any case we have added in the present version the effect of DMX on ICS staining for a better understanding of the effects of CBD on this model. AS described in the present version of figure 1, it is possible to observe that the effects of CBD are superimposable with the effects of DMX both on resting and activated ICS with the only exception of IL-4, in which the effects of DMX on activated cells was more pronounced that was observed with CBD. So far, even if the numerical differences are small, we are confident that the data reflect the real ability of CBD to affects ICS in this model. In addition, it is important to note, that ion the figure the data are presented as mean ±standard deviation and not standard error. A sentence in the result section was added to better explain these data and similar in the discussion section.

Same for L19/20 stating that Results show that CBD affects the frequency o of IFN- g/IL-17-producing cells . what is the biological significance of such an small change (e.g., 0.27% to 0.22%?) and with such a big deviation.

As above detailed, the presence of DMX in the model is suggesting that what we observe can reflect the ability of CBD to counteract activation-induced ICS content in CD4+ cells. The sentence was changed according to the data of the present version of the figure 1.

It is somehow confusing why does the authors  activate some times with PMA or Ionomycin, then with anticd3/Cd28, some times they use CD4+ cells and some times PBMCs,  some times the activation are in a period of 5 h  in other experiments 48 h… this must be clearly explained in other to compare or discuss the obtained results.

In the method section, in the paragraph “Isolation and proliferation assay of CD4+ T regulatory (Treg) and T effector (Teff)” the reason to use a different stimulus, for the different experimental models used is reported

The different time of incubation is used to assay different function's: 120 h for cell proliferation, while 48 h to measure cytokine protein production or mRNA expression of the different factors measured; in any caser this is better detailed in each method section’ in the paragraph named “Peripheral blood mononuclear cells (PBMCs) isolation and functional assays”

I personally found the data not clear and superficial, the authors must want to discussed or explore in more detail the mechanisms and implications of the observed effects.

The discussion section was improved and additional sentences were inserted to discuss this issue

Additionally:

Title: Peirpheral / nd

Error in the title was corrected

Keywords

it is highly encouraged to restrict the keywords to the TNF and IFN evaluated in this research.

Key words were simplified removing peripheral blood mononuclear cells, CD4+ T cells

22        Teff must be defined

In the relative method section “Isolation and proliferation assay of regulatory and T effector” CD4+ T cells

 Teff and Treg were better specified

23        Conclusions must be determined with clarity.  The authors must avoid the use of conditional terms like “may”  to conclude the effects of CBD in the cytokines production.  Additionally the data does not clearly support this conclusions (Table 1).

The conclusion was revised according to the results of the study

50        PMN must be defined

The full name was added and the acronymous is reported in parenthesis

73        2.1. Effect of CBD on T cells differentiation . This title does not correspond to the cells employed in this section.

The title was changed with “functional activation” and the two original subparagraphs remain the same

77/124  informal terms must be avoided in scientific texts. Please do not use contractions

77/78    and so on… % > percentage

79        figure> Figure, and must be bolded

80        PMA and IONO acronyms must be stated

From 77 to 80 changes were performed and lacking information’s added

82        only Figure number bolded, the rest must be normal text.

Titles were changed

Figure 1          in legend, concentrations of PMA and ionomycin must be stated here

Inserted

Figure 1          figure content must be indicated as a, b, c ,d  same for other figures.

Figure 1          please switch the column bars to scatter dot plot, it would be more informative. I personally consider it hard to appreciate the significance of the statistical approach employed with such a small change between conditions, and enormous deviations. Also, the statistical test must be stated in figure legend.

Figure was changed and letters identifying the different graphs were added; information’s about statists were added to the legend

Figure 1          for TH1-Th17 please clearly state that # is employed to compare ACT-CBD vs ACT alone. If it is the case is very hard to consider this data significantly different. Please state the statistical test employed.

Changed

Figure 2          Please state the statistical test employed.

Statistics test used was added for each figure and tables

Section 2.3     the data described here must be presented as figure or table with corresponding annotations (methodological, statistical…)

Section was changed and an additional tables (table 2a and 2b) describing these results were added

Figure sup. 1  what does stimulated mean? Authors must state clearly the methods employed.

A sentence clarifying this issue was added

201      citations must be homogenized to the journal format

Citations were revised and changed

216      0,1 > 0.1

numbers were changed according to the suggestions

Materials and methods  authors must include a section attending to the following instruction: when reporting on research that involves human subjects, human materialhuman tissues, or human data, authors must declare that the investigations were carried out following the rules of the Declaration of Helsinki of 1975 (https://www.wma.net/what-we-do/medical-ethics/declaration-of-helsinki/), revised in 2013. According to point 23 of this declaration, an approval from the local institutional review board (IRB) or other appropriate ethics committee must be obtained before undertaking the research to confirm the study meets national and international guidelines. As a minimum, a statement including the project identification code, date of approval, and name of the ethics committee or institutional review board must be stated in Section ‘Institutional Review Board Statement’ of the article

In the method section “Peripheral blood mononuclear cells isolation and functional assays” the motivation of lack of the Ethical Committee approval is now detailed: “Human Peripheral blood mononuclear cells (PBMCs) were obtained from unemployed blood buffy coats provided by the local blood bank (Ospedale di Circolo, Fondazione Macchi, Varese, Italy); these materials represents a discard of the whole blood for clinical use and therefore no Ethical Committee approval is requested”.

Materials and methods  the authors must clearly described the cells culture conditions. Media composition, etc.

Materials and methods.         It is allowed to briefly describe well-known methods, but it has to be appropriately cited, however in my opinion, the proposed citations employed by the authors (links) to explain the methods are not friendly or appropriated, as readers have to open at least 6 independent web pages to understand this section. Anyhow, this section should be described with sufficient detail to allow others to replicate and build on published results.

According to your suggestion a brief explanation, in addition to the web-method detail was added for each section

238      Technologies Italia) > Technologies, Italia)

Corrected

271      perm/whash

Corrected

336      CD4 positive cells (CD4+) it must be described since the first time of appearance, not here

Removed

344/345 is CD4+ meaning was defined previously it is not necessary to write CD4+ positive

Corrected

335      the authors must explain why is it necessary to abbreviate lymphocytes here? If it is not employed anywhere…

It was used to better explain the term Lympho that is present in the figure: now it was better explained

304      FSC and SSC the authors must state the meaning of this parameters, as the term is only informative for cytometrist.

In the legend a sentence to explain the terms was added

289 Suppl Table 2  > Suppl.

344 homogenize the table name to Suppl. Table 1 and homogenize font size or  type

Round 2

Reviewer 2 Report

In the  revised version the authors added dexamethasone as a positive control for ICC assays, however the approach only generated more questions.

How does the authors exclude the cytotoxic effects of DEX in such concentrations?

Modifications for cytokine evaluation (addition of dexamethasone) and comparison with the effect of CBD does not add any information, especially if dexamethasone is not tested alone.  

Additionally, why does the authors employ independent t-test for each condition? Considering the experimental approach and the presented data, was not an Analysis of Variance the recommended test? With corresponding multiple comparison test.  Same for the rest of the figures.

Figure labeling is recommended as a basic strategy to follow the manuscript content more easily, however the authors just attended to this suggestion superficially.  Only figure 1 was modified and the figure legend does not follow this format.

2.3 section does not reflect the effect of CBD on PBMC cell proliferation.  I do not doubt the effect of CBD in the proliferation of specific subsets from PBMCs, however CBD must be tested alone. And ACT with cd3 and cd 28 as positive control.  Additionally,  It is not clear if the P value was the employed for the analysis or if the

 L246 0,2  to 0.1 as in the rest of the MS

I respectfully disagree with the author’s conclusion stating that  CBD can profoundly affect their differentiation mRNA expression, activation, and cytokine content.

Author Response

Varese, august 30 2023

Editor-in-Chief: International Journal of Molecular Sciences

Dear Editor

            Enclosed please find the second revised version of the manuscript entitled “EFFECT OF CANNABIDIOL ON HUMAN PERIPHERAL BLOOD MONONUCLEAR CELLS AND CD4+ T CELLS

, by Alessia Furgiuele, et al.

The manuscript was revised according to the Reviewer’s suggestions and a point by point reply is reported hereafter. We are grateful to the Reviewers for their Comments and we agree that the revision has positively improved the quality of the manuscript.

All the changes have been highlighted in red throughout the text, and removed sentences or words are crossed out. Finally, the references list was revised according to the journal format; all the revised references are outlined in yellow and the additional references in red.

All authors approve the present version of the full-length manuscript, and hope that it can be nowaccepted for the publication in International Journal of Molecular Sciences.

All the data in this study are archived and accessible on reasonable request for review purposes by Editors and Reviewers.

Best regards

Marco Cosentino

Marco Cosentino

Full Professor in Medical Pharmacology

Center for Research in Medical Pharmacology

University of Insubria

Via Monte Generoso 71

21100 Varese VA - Italy

Phone +39 0332 217410

Fax +39 0332 217409

E-mail: marco.cosentino@uninsubria.it

Answers to reviewer comment’s

Reviewer 2

In the  revised version the authors added dexamethasone as a positive control for ICC assays, however the approach only generated more questions.

How does the authors exclude the cytotoxic effects of DEX in such concentrations?

In preliminary experiments, we tested DEX 1 microM for 5 h in our experimental model and found no evidence of cytotoxic or apoptotic effects. In the present version of the manuscript, we added a sentence in the results section to clarify this point. We chose this concentration based on published studies which investigate DMX effects on human isolated cells (for example, on human T lymphocytes: J Rheumatol. 2002 Jan;29(1):46-51).

Modifications for cytokine evaluation (addition of dexamethasone) and comparison with the effect of CBD does not add any information, especially if dexamethasone is not tested alone.

DMX has been added to compare CBD with an established anti-inflammatory drug. In our experiments, DMX was always tested alone, on resting as well as on stimulated cells. In Figure, we have now added also DMX alone, which was unable to affect resting conditions.

Additionally, why does the authors employ independent t-test for each condition? Considering the experimental approach and the presented data, was not an Analysis of Variance the recommended test? With corresponding multiple comparison test. Same for the rest of the figures.

We agree with the Reviewer that also analysis of variance is be a good statistical approach, but in this case. Nevertheless, our experimental design is always based on the comparison of only two conditions at a time (resting vs activated, resting alone vs resting + CBD, activated alone vs activated + CBD). An exception is now the set of experiments shown in Figure 1, where DMX was added in both resting and activated conditions. In this case, we have now applied One-way analysis of variance with Dunn’s post-test.

Figure labeling is recommended as a basic strategy to follow the manuscript content more easily, however the authors just attended to this suggestion superficially.  Only figure 1 was modified and the figure legend does not follow this format.

Figure legends were revised according to Reviewer’s suggestions.

2.3 section does not reflect the effect of CBD on PBMC cell proliferation.  I do not doubt the effect of CBD in the proliferation of specific subsets from PBMCs, however CBD must be tested alone. And ACT with cd3 and cd 28 as positive control.  Additionally,  It is not clear if the P value was the employed for the analysis or if the

Section 2.3 had the title “effect of CBD on cell proliferation” and includes the data on cell proliferation for both PBMC (lines: 143-146) that in the results are reported in table 2 (lines: 152-157), and for Teff/Treg subpopulation (lines: 147-149) that in result section are described in table 3 (lines: 158-164).

According to the data presented in figures 1 and 2, CBD was never able to affects resting cell function's; for this reason and in line with our primary aim of the section 2.3, that was to investigate if CBD can affects cell proliferation, considering that resting PBMC proliferation was (% of positive cells) 0.56±0.30 (of positive cells) and 0.55±0.51 for Teff, we don’t add the effects of CBD on the resting conditions of both experimental models. In any case, to answer to the reviewer question, in the present version the data were added in both tables and indicated as CBD on resting. Usually the model is performed for the analysis of proliferation under activating stimuli and the values of resting conditions are inserted only to underlines the entity of proliferation obtained with proliferating stimuli. The result section was modified, in view of better explain our aim.

 L246 0,2  to 0.1 as in the rest of the MS

Changed.

I respectfully disagree with the author’s conclusion stating that  CBD can profoundly affect their differentiation mRNA expression, activation, and cytokine content.

We modified the sentence and removed “profoundly affects”.

Round 3

Reviewer 2 Report

The authors have improved the quality of the manuscript presentation  across the review process.